# Tuning the Thermogelation and Rheology of Poly(2-Oxazoline)/Poly(2-Oxazine)s Based Thermosensitive Hydrogels for 3D Bioprinting

**DOI:** 10.3390/gels7030078

**Published:** 2021-06-24

**Authors:** Malik Salman Haider, Taufiq Ahmad, Mengshi Yang, Chen Hu, Lukas Hahn, Philipp Stahlhut, Jürgen Groll, Robert Luxenhofer

**Affiliations:** 1Functional Polymer Materials, Chair for Advanced Materials Synthesis, Institute for Functional Materials and Biofabrication, Department of Chemistry and Pharmacy, Julius-Maximilians-University Würzburg, Röntgenring 11, 97070 Würzburg, Germany; mengshi.yang@uni-wuerzburg.de (M.Y.); chen.hu@uni-wuerzburg.de (C.H.); lukas.hahn@uni-wuerzburg.de (L.H.); 2Department of Functional Materials in Medicine and Dentistry, Institute for Functional Materials and Biofabrication and Bavarian Polymer Institute, Julius-Maximilians-University Würzburg, Pleicherwall 2, 97070 Würzburg, Germany; taufiq.ahmad@fmz.uni-wuerzburg.de (T.A.); philipp.stahlhut@fmz.uni-wuerzburg.de (P.S.); juergen.groll@fmz.uni-wuerzburg.de (J.G.); 3Soft Matter Chemistry, Department of Chemistry and Helsinki Institute of Sustainability Science, Faculty of Science, University of Helsinki, PB 55, 00014 Helsinki, Finland

**Keywords:** poly(2-ethyl-2-oxazoline), shear thinning, shape fidelity, cyto-compatibility, bio-printability

## Abstract

As one kind of “smart” material, thermogelling polymers find applications in biofabrication, drug delivery and regenerative medicine. In this work, we report a thermosensitive poly(2-oxazoline)/poly(2-oxazine) based diblock copolymer comprising thermosensitive/moderately hydrophobic poly(2-*N*-propyl-2-oxazine) (pPrOzi) and thermosensitive/moderately hydrophilic poly(2-ethyl-2-oxazoline) (pEtOx). Hydrogels were only formed when block length exceeded certain length (≈100 repeat units). The tube inversion and rheological tests showed that the material has then a reversible sol-gel transition above 25 wt.% concentration. Rheological tests further revealed a gel strength around 3 kPa, high shear thinning property and rapid shear recovery after stress, which are highly desirable properties for extrusion based three-dimensional (3D) (bio) printing. Attributed to the rheology profile, well resolved printability and high stackability (with added laponite) was also possible. (Cryo) scanning electron microscopy exhibited a highly porous, interconnected, 3D network. The sol-state at lower temperatures (in ice bath) facilitated the homogeneous distribution of (fluorescently labelled) human adipose derived stem cells (hADSCs) in the hydrogel matrix. Post-printing live/dead assays revealed that the hADSCs encapsulated within the hydrogel remained viable (≈97%). This thermoreversible and (bio) printable hydrogel demonstrated promising properties for use in tissue engineering applications.

## 1. Introduction

Recent innovations in biomaterials have had a huge impact on all aspects of tissue engineering, regenerative medicine and drug delivery, resulting in development of “smart” biomaterials (responsive to external stimuli, e.g., temperature, pH, light etc.) [1,2]. Because of rapidly growing interests in precision medicine, emergence of gene/immune therapies and advancements in three-dimensional (3D) (bio) printing, there is an increasing demand for smart biomaterials.

A thermogel is a stimuli-responsive “smart” material which responds to change in temperature, above or below the critical temperature by a (typically reversible) sol-gel transition. The gelation process is typically due to the development of supramolecular structures yielding 3D, physically crosslinked networks because of hydrogen bonding, coulomb or hydrophobic interactions often in combination with simple entanglement [3]. In the majority of cases, the transition takes place upon increase in temperature [4] but inverse gelation has also been observed [5,6,7]. These are highly versatile materials and repetitively proposed for multiple applications in biomedical, pharmaceutical and food industry. Apart from thermogelling natural polymers [8], few synthetic polymers also exhibit this behaviour [9]. In this context, poly(*N*-isopropylacrylamide) [10,11] and few members of Pluronics^®^ family (also known as poloxamers) (ABA triblock copolymer where A is hydrophilic poly(ethylene glycol) (PEG) and B is thermoresponsive/hydrophobic poly(propylene glycol) (PPG)) [12] are most frequently discussed. Several studies have presented PEG as highly cytocompatible [13]. Furthermore, few members of the PEG family have been approved by US Food and Drug Administration as sealants [14,15]. However, the limited biodegradability [16,17,18] and more recently emerging, PEG immunogenicity [19] has remained the associated issues. A very interesting multiblock terpolymer from PEG, PPG and poly(ε-caprolactone) has been recently investigated for its utilization as a vitreous substitute [20].

Polymers of cyclic imino ethers, particularly poly(2-oxazoline)s (POx) and poly(2-oxazine)s (POzi) [21] with a thermoresponsive profile [22,23] have attained significant attention in the past decade [23,24]. POx has shown huge potential in tissue engineering [25,26,27,28,29], drug delivery [30,31,32,33,34,35] and 3D (bio) printing [24,36]. They are structural isomers of polypeptides, are synthetically relatively easily accessed and are highly tunable in their physico-chemical characteristics [37]. POx with fewer than four carbons in their side chain are water soluble. Regardless of temperature, poly(2-methyl-2-oxazoline) (pMeOx) is soluble in water while poly(2-ethyl-2-oxazoline) (pEtOx), poly(2-propyl-2-oxazoline) (pPrOx), poly(2-isopropyl-2-oxazoline) (p*i*PrOx) and poly(2-cyclopropyl-2-oxazoline) (p*c*PrOx) exhibit lower critical solution temperature (LCST) type behaviour and temperature dependent aqueous solubility [38,39]. Among these, pMeOx and pEtOx have been studied thoroughly as highly hydrophilic polymers which is discussed as one potential alternative to PEG, because of non-toxic and non-immunogenic nature, low unspecific organ uptake and rapid clearance (at right size) [25,40,41,42]. Utilizing the light scattering technique, Grube et al. explored the solution properties of POx in comparison to PEG, and reported that both pMeOx and pEtOx with same molar mass (as PEG) were less solvated and more compact in shape. These results further suggested that on basis of absolute physico-chemical properties, POx are an interesting alternative to PEG [43]. In various attempts to control the protein adsorption and cell adhesion on the biomaterials, the POx particularly pMeOx, pEtOx and pMeOzi (poly(2-methyl-2-oxazine)) showed better performance than PEG [44,45,46].

A plethora of studies show the utilization of pMeOx and pEtOx as hydrophilic polymer for the development of hydrogels [28,47,48,49], electrospun fibres [50], polymer brushes [51], nanoparticles [52], micelles for drug delivery [53,54,55,56,57] and for melt electrowriting [58,59]. Bloksma et al. also investigated the thermoresponsive behaviour of few POzi based homopolymers [22]. Interestingly though, there are few reports that solely address the thermogelation of pure POx/POzi based systems. Inspired by the Pluronics family, Zahoranova et al. studied various ABA and BAB triblock copolymers based on pMeOx (A) and pPrOx (B) but no thermogelation could be observed up-to 30 wt.% concentration at the investigated temperature range of 10 to 50 °C [60]. More recently, Lübtow et al. reported on the development of ABA triblock copolymer comprised of pMeOx (A) and poly(2-iso-butyl-2-oxazoline) (p*i*BuOx) (B) which form thermogels at 20 wt.% but its rheological characteristics, in particular its low yield stress were not conducive for extrusion-based 3D printing [48]. In addition, Hahn et al. reported on several ABA POx/POzi-based triblock copolymers exhibiting inverse gelation [6,61].

While both pMeOx and pEtOx (and more recently pMeOzi) are routinely considered as potential alternative to PEG or other hydrophilic polymers, these two polymers have shown distinct physico-chemical solution properties. Recently, we have observed that by replacing pMeOx with pEtOx as hydrophilic block (A) can dramatically reduce the drug loading efficiency of ABA type POx/POzi based triblock copolymer micelles [55]. Previously, we synthesized POx/POzi based thermogelling bio-ink composed of pMeOx (no LCST) as A and thermoresponsive poly(2-propyl-2-oxazine) (pPrOzi) (LCST ≈ 12 °C) [59] as B (pPrOzi-*b*-pMeOx). The aqueous solution of pPrOzi-*b*-pMeOx hydrogel exhibited reversible thermogelation above 20 wt.% concentration. This hydrogel proved to be cytocompatible and (bio) printable [47,62]. Here, we replaced pMeOx with pEtOx in this type of block copolymer and investigated how the rheological and thermogelling properties of the resultant diblock copolymer (pPrOzi-*b*-pEtOx) are affected. The morphology, cytocompatibility and (bio) printability of the hydrogel were further evaluated.

## 2. Results and Discussion

### 2.1. Synthesis, Characterization and Rheology

To date, few reports investigated the thermogelation of pure POx or POx/POzi based systems [6,47,48,60,63]. Very recently, Monnery and Hoogenboom reported a POx based BAB triblock copolymer with the degree of polymerization (DP) ranging from 50–100 and up to 1000 repeats units for pPrOx and pEtOx (B and A, respectively), with reversible thermal gelation. Hydrogel formation (above 20 wt.%) was only observed when the pEtOx had extremely high DP (i.e., pPrOx_100_-*b*-pEtOx_700_-*b*-pPrOx_100_) [63], in contrast for pPrOzi-*b*-pMeOx diblock, the gelation was observed at much lower DP (i.e., pPrOzi_50_-*b*-pMeOx_50_) [47]. It was further confirmed that the fine tuning of cloud point temperature (T_cp_) with respect to the block length is necessary for the development of thermoresponsive hydrogels [63]. Investigating the impact of polymer concentration on T_cp_ of various AB diblock copolymers, A being EtOx while switching the B to poly(2-heptyl-2-oxazoline) (HepOx), poly(2-butyl-2-oxazoline) (BuOx) or *i*PrOx, Hijazi et al. suggested that the copolymer with lower differences in their hydrophobicity might be best suited for development of thermoswitchable hydrogels [64,65].

In this regard, we have synthesized a thermoresponsive diblock copolymer which is based on thermosensitive pEtOx as hydrophilic (A) and pPrOzi more hydrophobic, yet still thermoresponsive block (B), respectively. It should be noted, as both pMeOx and pEtOx are highly water-soluble and well-known stealth-polymers, we were interested in investigating how the thermogelling/rheological properties and printibility of AB diblock copolymer would be affected when exchanging pMeOx with pEtOx.

Initially, we synthesized pPrOzi-*b*-pEtOx diblock copolymer with the DP = 50 for each block (i.e., pPrOzi_50_-*b*-pEtOx_50_). The polymer was analysed by ^1^H-NMR, GPC and DSC. The ^1^H-NMR spectra were in good agreement with targeted block lengths for individual blocks (Figure 1a) with a dispersity Ɖ ≈ 1.3. DSC thermogram showed a single distinct glass transition (T_g_) at 23 °C suggesting that no (micro)phase separation (Figure 1b) occurred in the solid. Previously, we had observed the similar behaviour for the POx/POzi based ABA triblock copolymers [66]. Tube inversion test is the simplest test to see if a self-supporting hydrogel is formed or not. This was performed with 25 and 30 wt.% pPrOzi_50_-*b*-pEtOx_50_ solution by keeping at room temperature and 37 °C (to simulate the body temperature) (Figure 1c). Both polymer solutions appeared as highly viscous liquids and no macroscopic gel was observed in vials as the material flowed under its own weight.

Whether the gel is formed or not was further studied by temperature dependent rheological measurements of 25 and 30 wt.% pPrOzi_50_-*b*-pEtOx_50_ polymer solution. Initially, dynamic oscillation temperature sweeps from 5 °C to 60 °C were carried out (Figure 1d). The gel point is usually defined as the crossover point of storage modulus (G’) and loss modulus (G”) and corresponding temperature as gel temperature (T_gel_). No gelation was observed in the temperature range of 5 °C to 37 °C. However, for both concentrations rapid increase in G’ and G” was observed at 23 °C and 12 °C, respectively. The increase in G” at 12 °C can be correlated to the T_cp_ of the pPrOzi block [22]. Additionally, at slightly higher temperatures, a T_gel_ was eventually observed at 39 °C and 42 °C for 25 and 30 wt.% polymer solution, respectively. However, the resulting gel was not very stable as G’ and G” start to decrease and the gel liquefied again at around 52 °C. In comparison, a 20 wt.% solution of pPrOzi_50_-*b*-pMeOx_50_ exhibited a sharp sol-gel transition and a G’ value of 4 kPa (at 27 °C) indicating the formation of relatively stiffer gel [47].

As previously discussed, the length of each block in block copolymers can significantly impact the gelation behaviour [63,64,65]. Accordingly, we synthesized a new diblock copolymer with chain length of hundred repeat units for each block, i.e., pPrOzi_100_-*b*-pEtOx_100_. The ^1^H-NMR spectra showed the desired block length (≈100) (Figure 2a) with dispersity Ɖ ≈ 1.4.

Previously, for the diblock copolymer pPrOzi_100_-*b*-pMeOx_100_, we had observed two T_g_ values, corresponding to the T_g_ values of the two homopolymers (pPrOzi ≈ 10 °C and pMeOx ≈ 77 °C) suggesting immiscibility and microphase separation of two individual blocks (Appendix A) [47,67]. In contrast, the DSC thermogram of pPrOzi_100_-*b*-pEtOx_100_ showed only a single T_g_ at ≈ 26 °C (slightly higher in comparison to pPrOzi_50_-*b*-pEtOx_50_ T_g_ ≈ 23 °C) suggesting that again no microphase separation (Figure 2b) occurred in the solid. This finding is particularly interesting as Hoogenboom recently reported that pEtOx and pPrOx homopolymers show immiscibility when the molar mass exceeded 10 kg/mol (i.e., DP ≈ 100) [68]. One might have expected that pEtOx and pPrOzi would be less miscible than pEtOx and pPrOx, but in the present case we have block copolymers at hand, while Hoogenboom and co-workers studied homopolymer blends, which can be expected to phase separate more readily. Thermogravimetric analysis (TGA) showed the commonly known good thermal stability of POx and POzi with major losses occurring ≥360 °C (Figure 2c).

After successful synthesis and characterization, the visual appearance of the aqueous solutions of pPrOzi_100_-*b*-pEtOx_100_ at various concentrations (1, 10, 15, 20, 25 and 30 wt.%) was noted. Up to 15 wt.%, all solutions were turbid at room temperature indicating the presence of mesoglobules or larger self-assemblies responsible for light scattering (Figure 3a; 1st row). For the homologue pPrOzi_100_-*b*-pMeOx_100_ diblock, dynamic and static light scattering suggested the formation of polymersomes [69] while, for pPrOzi_100_-*b*-pEtOx_100_ diblock, at this point, we can only speculate the presence of similar assemblies. In contrast, in ice cold water, all the samples were optically transparent. This can be attributed to the T_cp_ of pPrOzi block (13 °C as homopolymer) [22]. Similar observations were made for pPrOzi-*b*-pMeOx based diblock copolymer (turbid below 10 wt.%) [47]. Above 15 wt.% all the tested concentrations were optically clear, at 20 wt.% the solution appeared as highly viscous yet flowable liquid. Self supporting, optically clear hydrogels were observed at 25 and 30 wt.% polymer solution at room temperature (Figure 3a; 2nd row).

In contrast to pMeOx, pEtOx also exhibits thermoresponsive solubility behavior in water. Lin and colleagues reported T_cp_ values in water ranging between 61–69 °C, which were highly dependent on molar mass and concentration [70]. According to Hoogenboom, Schubert and colleagues, pEtOx homopolymers with degree of polymerization less than hundred do not exhibit T_cp_ (below 100 °C) but a chain length dependent decrease in T_cp_ (94 to 66 °C) was observed when chain length is greater than hundred [37,71]. Accordingly, upon heating the (25 wt.%) pPrOzi_100_-*b*-pEtOx_100_ hydrogel to 70 °C, it immediately turned opaque which was reversed upon cooling (Figure 3b). Throughout heating and cooling, the gel character was maintained, i.e., no syneresis was observed. This opacity can be correlated to the T_cp_ and dehydration of pEtOx at this temperature. It can be expected that the T_cp_ of pEtOx is lower in the block copolymer compared to the homopolymer [72].

Rheological properties of solutions of 15 wt.% to 30 wt.% were investigated in dependence of temperature (5 to 60 °C). After gelation, G’ reached a plateau at values of 3 (at 30 °C) and 1.5 kPa (at 25 °C) for 30 and 25 wt.% polymer solution, respectively (Figure 4a). While many other thermogelling polymers solutions from literature are weaker (G’ < 1 kPa) [73,74], the homologue 20 wt.% pPrOzi_50_-*b*-pMeOx_50_ based hydrogel is in fact somewhat stronger (G’ ≈ 4 kPa) [47]. The loss factor (tan δ = G”/G’) informs on the ratio of elastic vs. viscous property of the material (tan δ < 1, more elastic and tan δ > 1, more viscous). At both concentrations (30 and 25 wt.%), rather low tan δ values of ≈ 0.07 and 0.16, respectively, were obtained. Accordingly, once formed, the pEtOx based hydrogels show a slightly higher elastic character compared to those bearing a pMeOx hydrophilic block (tan δ ≈ 0.2) [47].

The crossover point for G’ and G” was also observed for 20 and 15 wt.% polymer solution at 35 and 41 °C, respectively. However, the resultant gel at such concentration was very weak (G’ < 0.2 kPa). The G’ value of the 15 wt.% polymer solution registered as zero until 40 °C but increased rapidly above this temperature. Overall, the temperature sweep (5 to 60 °C) showed an inverse relationship between T_gel_ and polymer concentration, i.e., for 30, 25, 20 and 15 wt.% polymer solutions, the obtained T_gel_ was 13, 15, 35 and 41 °C (Figure 4b). In case of pPrOzi-*b*-pEtOx, increasing DP from 50 to 100 resulted in a stable hydrogel, while in case of pPrOzi-*b*-pMeOx based hydrogels, without a significant change in G’ values, the T_gel_ was decreased from 27 to 17 °C, respectively [47].

To assess the linear viscoelastic (LVE) region (stable values for G’ and G”), an amplitude sweep was conducted with constant angular frequency of 10 rad/s and increasing amplitude of 0.01 to 500% (Figure 5a). Within the LVE range, this test confirmed the occurrence of only elastic rather than plastic deformations (which can lead to damage of sample). With G’ > G” for 30 and 25 wt.% polymer samples, the viscoelastic solid character of the material is confirmed. The sharp drop in G’ indicate a brittle fracturing behaviour. Before this occurs, the G” increases before reaching a peak maximum, after which the curve again dropped. The common interpretation of the rise in G” in zone 1 (exemplified with 30 wt.% polymer solution, black dashed lines) is the occurrence of micro-cracks while the overall structural integrity is maintained as G’ remains constant and dominant over G” (Figure 5a) [75,76]. However, for the present materials, one might rather assume that small domains start to liquefy under the increasing stress, increasing the loss modulus but retaining the storage modulus until further breakdown of the network. With further increasing amplitude (zone 2) individual domains grew further and connect, resulting in the flow of entire material (G” > G’: fluid state). The 20 wt.% solution showed minor differences in G’ and G” values but at very low amplitude sweep, a crossover point was observed which is indicative of fluid state, while G”> G’, in case of 15 wt.% confirmed the free-flowing liquid (Appendix A). The yield stress (or linearity limit) is defined as the value of shear stress at the limit of LVE region, while flow point is represented as the value of shear stress at the crossover point of G’ and G” (where, G’ = G”) [77]. In the region between yield stress and flow point where G’ is still greater than G”, the material has already lost the initial structural strength but still displays the properties of solid material. Further increase in shear stress (above flow point) eventually leads to a flow of material (G” > G’). In (bio) printing, these are crucial factors which are usually correlated to the collapse and shape fidelity of structures being printed [78,79]. The yield stress and flow point values of 30 and 25 wt.% pPrOzi_100_-*b*-pEtOx_100_ hydrogels were extracted from amplitude sweep measurements (Figure 5b). Increasing the polymer concentration from 25 to 30 wt.% led to an increase in yield stress (from 110 to 310 Pa) and flow point values (from 160 to 520 Pa), respectively. A frequency sweep revealed little frequency dependency in the investigated range (0.1–100 rad/s) within the LVE range and 0.1% strain. Only at lowest frequencies and 25 wt.%, the tan δ approached unity (Figure 5c).

When extrusion-based applications such as 3D (bio) printing and injectable drug depots are desired, shear thinning, and fast structure recovery are important factors to assess. For pPrOzi_100_-*b*-pEtOx_100_ hydrogels the viscosity decreased profoundly as shear rate increased (0.01 to 1000, 1/s) at both 30 and 25 wt.% (Figure 5d). The flow index *n* in the power-law expression described by Ostwald-de Waele (*n* < 1 indicate shear thinning) and consistency index K characterize the gels [80]. The presently studied materials can be clearly modeled using the power-law expression using very low flow and rather high consistency indices (*n* ≈ 0.16 and 0.27, K ≈ 380 and 152, for 30 and 25 wt.% hydrogel, respectively) (Figure 5d; dashed lines). The structure recovery was investigated using a rotational approach with an angular frequency of 10 rad/s and temperature of 25 °C (Figure 5e). At first, a low amplitude strain regime of 0.1/s was applied for 100 s followed by higher amplitude strain of 100/s for 100 s (5 cycles). At both tested concentrations, a sharp decrease in viscosity at high shear rate was recovered instantly upon cessation of shear rate. Together, pronounced shear thinning and rapid recovery facilitate extrusion through fine nozzles with shape fidelity after leaving the nozzle, which is relevant for 3D printing of bio-ink or for drug delivery applications.

### 2.2. SEM and Cryo-SEM Analysis

For tissue engineering applications, the printed scaffolds should provide a 3D environment with sufficient porosity to ensure nutrient and oxygen supply to the encapsulated cells. To obtain insights into the morphology and microarchitecture of hydrogels, initially, the images of the 30 wt.% pPrOzi_100_-*b*-pEtOx_100_ lyophilized gel were obtained by SEM. The lyophilized gel exhibited a highly porous, rather heterogeneously distributed network of capillary channel (Figure 6a) and the top view showed randomly distributed micropores, probably produced because of water sublimation (Appendix A). As currently discussed in scientific community, factors such as pore formation by water evaporation during lyophilization and/or ice crystal formation during slow freezing can strongly influence appearance of a freeze-dried hydrogels and might cause artifacts [81]. To elude such possibilities, the cryogenic SEM (cryo-SEM) was also conducted. During the sample preparation for cryo-SEM, the slush nitrogen provided the sufficient cooling rates resulting in faster cooling and reduction of crystallization artifacts. The cryo-SEM revealed that the intrinsic hydrogel morphology was preserved and pPrOzi_100_-*b*-pEtOx_100_ hydrogel exhibited highly homogenous, honeycomb-like porous structure in the sub-micrometer range (Figure 6b). Generally, the high porosity is significant for drug delivery and biomedical applications due to its capacity to facilitate drug loading and release, also offering void space for oxygen and nutrient transport for cell proliferation.

### 2.3. 3D-Printing of Hydrogels

The printability of pPrOzi_100_-*b*-pEtOx_100_ based hydrogel was assessed using extrusion-based 3D printing. As this polymer formed hydrogel at/above 25 wt.%, the printing experiments were performed at 25 and 30 wt.% polymer concentrations. The filament fusion test was accomplished to assess the shape fidelity or print accuracy, strand fusion and undesired spreading of the printed construct. The shape fidelity or print accuracy can generally be defined as the degree of dimensional faithfulness of the printed object in comparison to computer aided design [82] and in the case of pPrOzi_100_-*b*-pEtOx_100_ based hydrogels, the assessment was accomplished only qualitatively, by visual inspection. A 2D pattern of one meandering filament with stepwise increasing strand distances from 0.5 to 0.75, 1, 1.25 and 1.5 mm was printed (Figure 7a). Even at the lowest distance (0.5 mm), the printed filament did fuse only at a few points but mainly remained separated, above this distance, the filaments were fully separated. To assess filament stability and sagging under its own weight when suspended, a filament collapse test was performed on a serrated mold with stepwise increasing gaps of 1, 2, 4, 8 and 16 mm. At 30 wt.% concentration, the suspended hydrogel filament strand did show some sagging but remained continuous even at the longest distance of 16 mm (Figure 7b), while at 25 wt.% the filament collapsed in the case of the 16 mm gap.

The printability and shape fidelity were further assured by printing various shapes such as a serpentine line pattern, five-pointed star and a 2-layered 5 × 5 strand grid (Figure 8a). The sharp filament corners (in serpentine such as line and star) and intersections in the grid were all well resolved. In the case of pPrOzi-*b*-pMeOx based diblock copolymer, Hu et al. showed that such kind of printability and shape fidelity was achieved by the addition of clay (Laponite XLG 1.2–2 wt.%) while the polymer alone exhibited a lower quality printing, albeit at 20 wt.% diblock concentration [67].

### 2.4. Printability and Rheology of POx-Laponite Hybrid Hydrogels

In fact, laponite is routinely applied as rheology modifier [83,84] and has shown potential in biomedical applications [85]. Laponite comprises disc-like silicate particles. It is non-toxic and capable to enhance the biological activities such as cell adhesion and proliferation [86,87]. As a cyto-compatible additive, it can not only improve mechanical strength and printability [88] of the bio-ink but also biological effects [89]. The development of multilayered successive alternating structure (high stackability) is one of the most desirable characteristics for (bio) printing of the constructs for tissue engineering applications [90]. In order to observe the stackability, the pristine hydrogel (25 wt.% pPrOzi_100_-*b*-pEtOx_100_) and a hybrid hydrogel (with 1 wt.% additional laponite XLG) were printed. For the plain hydrogel, printing more then 3 layers greater lead to slight flattening of the scaffold (Appendix A) while with only 1 wt.% laponite XLG a 12 layered star was easily printed with good shape fidelity and no collapse of the high single filament stacks (Figure 8b). Cryo-SEM of the hybrid hydrogel revealed the uniform microporous structure and pore size of the hydrogel (Appendix A) is apparently not affected by the presence of the added laponite.

Rheology of a material give generally a good indication for printability [91]. Similar to many other polymers [67,85,88,92], in this particular case, the laponite also improved the printability (in terms of stackability) of pEtOx based hydrogel. To correlate the rheology with printability, the rheological profile of hybrid hydrogel (under same set of conditions as previously explained) was obtained and compared to plain hydrogel. No significant difference was observed in T_gel_, while the storage modulus increased from 1.5 kPa for plain hydrogel to 3 kPa for hybrid (at 25 °C), indicating a stiffer gel, which may be attributed to the physical interaction between polymer and laponite (Appendix A). The loss factor tan δ decreased from 0.16 (plain) to 0.05 (hybrid) indicating a higher elastic character. No significant differences were observed in overall LVE range of both plain and hybrid hydrogel. However, the yield point values for hybrid hydrogel (190 Pa) was found to be higher than plain hydrogel (110 Pa) (Appendix A) which is likely to be one important factor responsible for improved stackability. The dynamic frequency sweep experiment revealed that G’ value of hybrid hydrogel remained stable and higher than G” value throughout the investigated frequency range indicative of typical gel phase (Appendix A) while the plain hydrogel appeared to be in transition phase of gelation at low shear frequency. In comparison to the plain hydrogel, the steady state flow test revealed more pronounced shear thinning behaviour for hybrid hydrogel (Appendix A) which is particularly interesting and beneficial for (bio) printing, as cells are more prone to mechanical disruption caused by extrusion pressure from the printing cartridge. Shear thinning is beneficial to reduce shear stress (if the yield points are sufficiently low). A structure recovery test was also performed to investigate the effect of clay on reversible sol-gel transition under episodes of high and low strain values (Appendix A). Cyclic strain sweeps revealed that at high and low strains, the hybrid hydrogel was capable of transitioning from predominantly elastic to viscous material.

In summary, the addition of small amount of clay significantly improved the rheological properties of pPrOzi_100_-*b*-pEtOx_100_ hydrogels (and in turn printability) without compromising the innate thermogelling behaviour. In most cases, the interaction of clay with polymers is indirectly explored rather than directly investigating the interactions at molecular levels. Very recently, Le Coeur et al. studied the interactions of clay with pMeOx and PEG by NMR spectroscopy and small angle neutron scattering [92]. The results revealed that pMeOx had stronger interaction with clay than PEG. At this point, we have only investigated a single combination of hybrid hydrogel, which showed promising results. By further tuning the polymer and clay ratio, it may be possible for further tailoring the rheological properties. Additionally, it will be very interesting to study in depth the effects of addition of laponite to pPrOzi-b-pMeOx and pPrOzi-b-pEtOx based hydrogels, as the potentially different interactions between the polymers and the clay nanoparticles may affect the rheological properties quantitatively different. Very recently we have observed that pPrOzi-b-pMeOx/clay combination has proved to be highly efficient fugitive support material [93] for poorly printable sodium alginate [94]. We anticipate that pPrOzi-b-pEtOx/clay combination also holds similar potential and can be explored as fugitive support material for variety of polymers. However, such detailed investigation of polymer/clay interactions and application of pPrOzi-b-pEtOx as fugitive material are separate studies at their own and are beyond the scope of current contribution.

### 2.5. Biological Studies

In addition to the seeded cells, the scaffold providing a 3D micro-environment to the cells is an essential element in tissue engineering. An appropriate scaffold should provide a room for homogeneous distribution of cells, should be cyto-compatible, non-toxic and promote the cell’s proliferation and differentiation. The generally excellent cyto-compatibility of POx-based polymers has been repeatedly shown while POzi-based polymers were studied to a lesser extent [24,36,66,95]. The cyto-compatibility of a certain polymer class must be assessed based on a defined set of experiments, cell-type and should never be generalized but rather verified on case-by-case basis.

Herein, we investigated the cell distribution and the impact of printing process on the cell survival in this new material. The 30 wt.% pPrOzi_100_-*b*-pEtOx_100_ hydrogels used for biological studies were prepared in cell culture medium. Initially, the rheological properties of the hydrogels prepared in media were investigated to screen for additional effect of media. Generally, the cell culture media did not significantly impact the rheological properties (Appendix A), except a slight decrease in gel strength was observed (i.e., from 3 kPa to 2 kPa) when compared to water-based hydrogels, albeit the same handling and printing conditions were used. This highlights to benefit of using a non-ionic bio-ink, as it is less prone to be affected by physiologically relevant proteins, salt concentrations and pH values.

Here, we used human adipose derived stem cells (hADSCs), which have gained wide attention in bioprinting, not only to show the cyto-compatibility of developed material for a somewhat more sensitive cell line, but also to pave way to future tissue engineering applications, where stem cells will likely be preferably used. The fluorescently labelled hADSCs were mixed in the hydrogels (for details, see Section 4) followed by printing and imaging. Important to note, the printability was not influenced by the presence of cells and shape fidelity was entirely preserved.

The homogenous cell distribution was observed throughout the printed construct (Figure 9a). The low viscosity of thermoresponsive hydrogel facilitated the cell distribution just by gently mixing at low temperature. The cell laden bio-ink was further visualized with cryo-SEM to observe the lodging of hADSCs in the hydrogel and to observe (if any) change in overall morphology of hydrogel itself because of cell culture media. Cryo-SEM of the cell laden bio-ink revealed a random distribution of the cells within hydrogel matrix showing cell interior with various organelles (Figure 9b) and no apparent change in porous structure of the hydrogel was observed. The images also make very clear that the pores of the hydrogel are much smaller than the cell diameter.

The cytocompatibility of POx based amphiphiles have been established several times [42,47,95]. In addition to the homogenous distribution of cells in the hydrogels, the measurement of cell viability in the printed scaffolds is more relevant. There are multiple factors such as printing pressure, nozzle inner diameter and shear profile of hydrogel etc. that can directly affect the cells viability and proliferation. The cell mechanical disruption is the direct consequence of pressure gradient/shear stress produced during bioink extrusion from the nozzle. In this regard, post printing live and dead assay was performed to evaluate the proportion of living (green) and dead (red) cells. Two constructs (serpentine line and a star) were printed and immediately visualized under fluorescence microscope (Figure 10a,b). A high extent of viable cells (97 and 98% in serpentine line and a star shape, respectively) confirmed that the printing parameters did not affect the cell viability. For better visibility, the few dead cells (red) in each individual region are highlighted in Appendix A.

## 3. Conclusions

In summary, thermosensitive pPrOzi_100_-*b*-pEtOx_100_ based diblock copolymer was successfully synthesized by living cationic ring opening polymerization and was further characterized by ^1^H-NMR spectroscopy, GPC, DSC and TGA. The pPrOzi_100_-*b*-pEtOx_100_ aqueous solution (i.e., 30 wt.%) underwent reversible sol-gel transition at T_gel_ around 13 °C. Stable and optically clear hydrogels with microporous structure and relatively high mechanical strength of G’ ≈ 3 kPa (at 30 wt.%), were obtained. At the same time this material is highly shear thinning with good structure recovery properties. Additionally, with extrusion-based 3D printing technique, various 2D and 3D patterns with high resolution and shape fidelity could be printed. The hADSC stem cells could be efficiently encapsulated into the hydrogels and this material appeared to be cyto-compatible in post-printing cell experiments.

The newly synthesized diblock pPrOzi-*b*-pEtOx as a structural variant to a previously reported pPrOzi-*b*-pMeOx polymer has potential for further exploration. However, pMeOx and pEtOx are routinely addressed as alternative and similar material [39,40,83] when compared to PEG but it will be interesting to see more of direct comparison of both polymers at molecular level. Advanced applications in biofabrication still may benefit from new smart materials which should be tunable with respect to physicochemical and rheological properties, (bio) printability and cyto-compatibility. With this diblock copolymer, an alternative platform which satisfies most demands needed in the context of biofabrication is introduced. We anticipate that the highly tunable character of POx with respect to rich chemistry would enable us to add crosslinking and/or bio (functionalization), albeit that the range of applications will be affected by the relatively high necessary concentrations. Considering this, the present materials could be of particular interest as a fugitive additive to be combined with other bioinks, to improve their printability or as a drug delivery platform.

## 4. Material and Methods

### 4.1. Materials

All substances for the preparation of the polymers were purchased from Sigma-Aldrich (Steinheim, Germany) or Acros (Geel, Belgium) and were used as received unless otherwise stated. The monomers 2-*N*-propyl-2-oxazine (PrOzi) and 2-ethyl-2-oxazoline (EtOx) were prepared following the procedure by Witte and Seeliger [96]. All substances used for polymerization, specifically methyl trifluoromethylsulfonate (MeOTf), EtOx, PrOzi and benzonitrile (PhCN), were refluxed over calcium hydride or phosphorus pentoxide and distilled and stored under argon. Laponite XLG was purchased from BYK-chemical GmbH (Wesel, Germany). Human adipose-dervived stem cells (hADSCs) were purchased from Lonza (Basel, Switzerland). Dulbecco’s modified eagle medium (DMEM) + GlutaMAX and Fetal bovine serum (FBS) were obtained from Gibco (Darmstadt, Germany) and Penicillin and streptomycin (P/s) solution were purchased from Biochrom AG (Berlin, Germany).

### 4.2. Methods

#### 4.2.1. Diblock Copolymer Synthesis

The polymerization and workup procedures were carried out as described previously [47,96]. As a general synthetic procedure, the initiator MeOTf was added to a dried and argon flushed flask and dissolved in the respective amount of solvent (PhCN). The monomer PrOzi was added to the reaction mixture and heated to 120 °C for approximately 12 h. Reaction progress was controlled by ^1^H-NMR-spectroscopy. After complete consumption of PrOzi monomer, the mixture was cooled to room temperature and the monomer for the second block, i.e., EtOx was added. The reaction mixture was heated to 110 °C for 4–6 h. After complete monomer consumption was confirmed, termination was carried out by the addition of 1-Boc-piperazine (PipBoc) at 50 °C and kept on stirring for 6 h. Subsequently, potassium carbonate was added and the mixture was stirred at 50 °C for further 4 h. The solvent was removed at reduced pressure under schlenk line. The dried polymer mass was dissolved in deionized (DI) water in ice, followed by process of rotary evaporation resulting in concentrated polymer solution. The polymer solution was transferred to a dialysis bag (MWCO 10 kDa, cellulose acetate) and dialyzed against DI water for 48 h. The polymer solution was recovered from the dialysis bag and lyophilized, finally obtained as white powder.

#### 4.2.2. Nuclear Magnetic Resonance

Deuterated solvents for NMR analysis were obtained from Deutero GmbH (Kastellaun, Germany). NMR spectra were recorded on a Fourier 300 (300.12 MHz), Bruker Biospin (Rheinstetten, Germany) at 298 K. The spectra were calibrated to the signal of residual protonated solvent (CDCl_3_ at 7.26 ppm). All the data were analysed by using the MNova software.

#### 4.2.3. Gel Permeation Chromatography (GPC)

GPC was performed on an Agilent 1260 Infinity System, Polymer Standard Service (Mainz, Germany) with Hexafluoro isopropanol (HFIP) containing 3 g/L potassium trifluoroacetate; having precolumn: PSS PFG 7µm, 5 cm length, 0.8 cm diameter followed by one AppliChrom ABOA HFIP-P-350 30 cm length, 0.8 cm diameter main column. The columns were kept at 40 °C and flow rate was 0.3 mL/min (HFIP). Prior to each measurement, samples were dissolved in HFIP and filtered through 0.2 μm PTFE filters, Roth (Karlsruhe, Germany). Conventional calibration was performed with PEG standards (0.1–1000 kg/mol) and data was processed with Win-GPC software.

#### 4.2.4. Dialysis

The synthesized polymer was dialysed by using Spectra/Por membranes with a molecular weight cut-off (MWCO) of 10 kDa (material: cellulose acetate) obtained from neoLab (Heidelberg, Germany). Deionized water (DI) water was renewed after 1 h, 4 h and every 12 h subsequently, until end of dialysis (48 h).

#### 4.2.5. Differential Scanning Calorimetry (DSC)

DSC was performed on DSC 204 F1 Phoenix equipped with a CC200 F1 Controller, (NETZSCH, Selb, Germany). The dynamic scans were recorded in nitrogen atmosphere with a heating rate of 10 K/min (25–200 °C) and subsequently cooled to −50 °C (10 K/min). The samples were heated and cooled two additional times from −50 °C to 200 °C (10 K/min) (three heating and 2 cooling cycles). For DSC studies, samples (8 to 10 mg) were placed into flat-bottom aluminum pans with crimped-on lids (pierced on the top).

#### 4.2.6. Thermogravimetric Analysis (TGA)

Thermogravimetric analysis was performed on TG 209 F1 IRIS, NETZSCH (Selb, Germany). The powdered polymer samples (10–15 mg) were placed in aluminium oxide crucibles (NETZSCH Selb, Germany) and heated under synthetic air from 30 °C to 900 °C with the heating rate of 10 K/min while detecting the mass loss.

#### 4.2.7. Rheology Measurements

Rheological measurements were performed using the MCR 301 rheometer from Anton Paar (Ostfildern, Germany) using a 25 mm diameter parallel-plate geometry and a Peltier system for temperature control. First, under a constant angular frequency and strain of 10 rad/s and 0.5%, respectively, the temperature sweeps from 5 to 60 °C were carried out at a heating rate of 0.05 °C/s to study the thermogelling behavior. Second, amplitude sweeps in the oscillation strain range of 0.01–500% was performed at a constant angular frequency of 10 rad/s, from which the linear viscoelastic (LVE) range was determined. Third, the frequency sweep was performed in an angular frequency range of 0.1–500 rad/s at a certain strain of 0.5% (within the LVE range obtained from the amplitude sweep). Finally, the steady-state shear flow from 0.01 to 1000/s of shear rate were performed to characterize the shear thinning behavior. The structure recovery properties were investigated utilizing the rotational approach with alternating (5 cycles) low (0.1/s) and high (100/s) amplitude strain (100 s each). Except for the temperature sweep, all tests were performed at 25 °C. A solvent trap was utilized in all experiments to prevent drying.

#### 4.2.8. Scanning Electron Microscopy

The 30 wt.% pPrOzi_100_-*b*-pEtOx_100_ hydrogel was frozen with liquid nitrogen and lyophilized afterwards. The lyophilized hydrogel was mounted on aluminum sample holders with conductive carbon tape and sputtered with a 4 nm layer of platinum in a sputter coater (Leica Microsystems ACE 400, Wetzlar, Germany). The morphology of the samples was subsequently analyzed using a Crossbeam 340 field emission scanning electron microscope (Carl Zeiss Microscopy, Oberkochen, Germany) by setting the acceleration voltage ETH to 2 kV, and detection of secondary electrons (SE) was performed with an Everhart-Thornley detector. The obtained images were further processed in imageJ software (1.46 r, revised edition).

#### 4.2.9. Cryo-Scanning Electron Microscopy

The 30 wt.% pPrOzi_100_- *b*-pEtOx_100_ hydrogel was also visualized with cryo-SEM. Samples were rapidly frozen in slushed nitrogen at −210 °C after placing them between aluminum plates (d = 3 mm) with a 2 mm notch for sample fixation. All the following transfer steps were performed at −140 °C with a EM VCT100 cryo-shuttle (Leica Microsystems). To generate a freshly fractured hydrogel surface, one of the aluminum plates was knocked off and freeze etched for 15 min at −85 °C under high vacuum (<1 × 103 mbar) in a Sputter Coater machine (ACE 400, Leica Microsystems). Afterward, samples were sputtered with 3 nm platinum and transferred to the SEM chamber (Crossbeam 340, Zeiss). Images of the hydrogel surface morphology were taken at −140 °C using an acceleration voltage of 2–8 kV. The images were further processed in ImageJ software (1.46 r, revised edition). The same process was also applied for cell-laden hydrogels.

#### 4.2.10. D-Printing of Hydrogel

Hydrogel printing experiments were performed using the extrusion-based 3D bioprinter BIO X from CELLINK (Gothenburg, Sweden) equipped with pneumatic driven print head and a 0.25 mm inner diameter precision needle (25 G). The extrusion pressure and printing speed was controlled and varied according to different user-defined printing structures, which were programmed by G-code. The ice-cold pPrOzi_100_-*b*-pEtOx_100_ solution was loaded as the ink into a printing cartridge and kept in 4 °C fridge before printing to eliminate air bubbles. The pPrOzi_100_-*b*-pEtOx_100_ filled printing cartridge was mounted in the 3D printer and driven pneumatically through the nozzle to print the hydrogel on a petri dish placed on 37 °C print-bed.

#### 4.2.11. Biological Studies

##### a. Cell Culture

For biological evaluation, human adipose derived stems cells (hADSCs) were cultured in DMEM + GlutaMAX supplemented with 10% FBS and 5% P/S under standard culture conditions (37 °C and 5% CO_2_). At 80 to 90% confluence, the cells were rinsed with phosphate buffer saline (PBS pH 7.4) and passaged using trypsin as detachment agent. The cell suspension was centrifuged for 5 min and the cell pellet was resuspended in culture media. The exact number of cells were obtained by manual cell counting.

##### b. Polymer Sterilization and Bio-Ink Preparation

Prior to each cell experiment, the weighed amount of polymer was sterilized under UV light (254 nm) for 1 h followed by addition of cell culture media. The polymer dispersion was kept in fridge at 4 °C until a clear solution was obtained. The cell suspension was added to the liquefied hydrogel kept in ice bath followed by 10 min of continuous slow manual rotation for homogenous distribution of cells (avoid using vortex mixer, it can damage the cells). The cell count was kept at 0.5 million cells/mL of 30 wt.% hydrogel for all experiments unless otherwise stated. The cell laden bio-ink was transferred to the printing cartridge and used accordingly.

##### c. Cell Distribution

To observe the cell distribution in hydrogel, the human adipose derived stem cells (hADSCs) were initially labelled with green fluorescent dye (cell tracker Green CMFDA from Invitrogen). In brief, the harvested cell pellet (1 million cells/mL) was resuspended with fluorescent reagent and incubated for 30 min in serum free media according to the manufacturer’s instructions. After centrifugation, the labelling solution was removed followed by addition of fresh pre-warmed media and cells were incubated for additional 30 min. The labelled hADSCs cells were dispersed in the prepared hydrogel as explained earlier. The serpentine line-like construct was printed, visualized and photographed under fluorescent microscope (Zeiss Axioimager Z1 microscope) at λ_Ex_ and λ_Em_ wavelength of 492 and 517 nm, respectively.

##### d. Post-Printing Live and Dead Assay

The Live/Dead viability assay was used to assess cell viability in the printed scaffolds according to the manufacturer’s instructions. Initially the harvested hADSCs pellet was dispersed in live/dead reagent solution prepared in cell culture media (calcein-AM (1:1000) and ethidium bromide (1:500), respectively). After 5 min of incubation, the cell suspension was added to the ice-cold liquid hydrogel and mixed (as explained earlier). In brief, 2 constructs with cell laden bio-ink were printed and samples were immediately examined and photographed by a fluorescent microscope (Zeiss Axioimager Z1 microscope). The live and dead cells appeared as green and red at excitation/emission wave length of 492/517 and 528/617 nm, respectively. The cell viability was reported as the ratio of the number of live cells to the total number of cells in each image counted with ImageJ software (1.46 r, revised edition). We estimate that from first mixing of the bio-ink to the live/dead analysis, the cells were exposed roughly one hour to the bio-ink. From each scaffold (*n* = 1), live cells with green fluorescence in cytosol or dead cells with red fluorescence in nuclei were counted at 4 different locations with in each construct.

## Figures and Tables

**Figure 1 gels-07-00078-f001:**
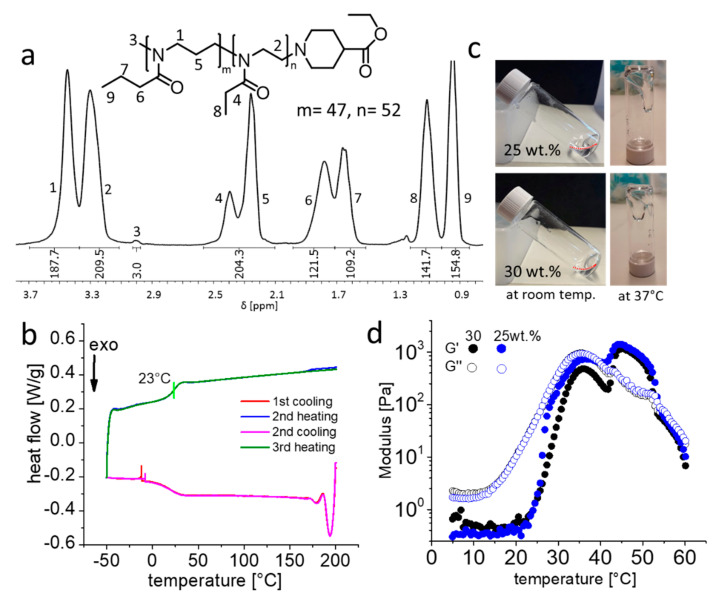
(**a**) Chemical structure and ^1^H-NMR spectra (CDCl_3_, 300 MHz, 298K) of pPrOzi_50_-*b*-pEtOx_50_ with the signal assignment of all major peaks. (**b**) Heat flow during several heating and cooling cycles (10 K/min) of differential scanning calorimetry, green vertical line indicating the glass transition point. (**c**) Visual appearance of the 25 and 30 wt.% polymer solution at room temperature and 37 °C, the red dotted lines are showing the liquid meniscus. (**d**) Temperature dependent rheology from 5 to 60 °C (at heating rate of 0.05 °C/s) of the 25 and 30 wt.% block copolymer solution with storage modulus (G’) and loss modulus (G”).

**Figure 2 gels-07-00078-f002:**
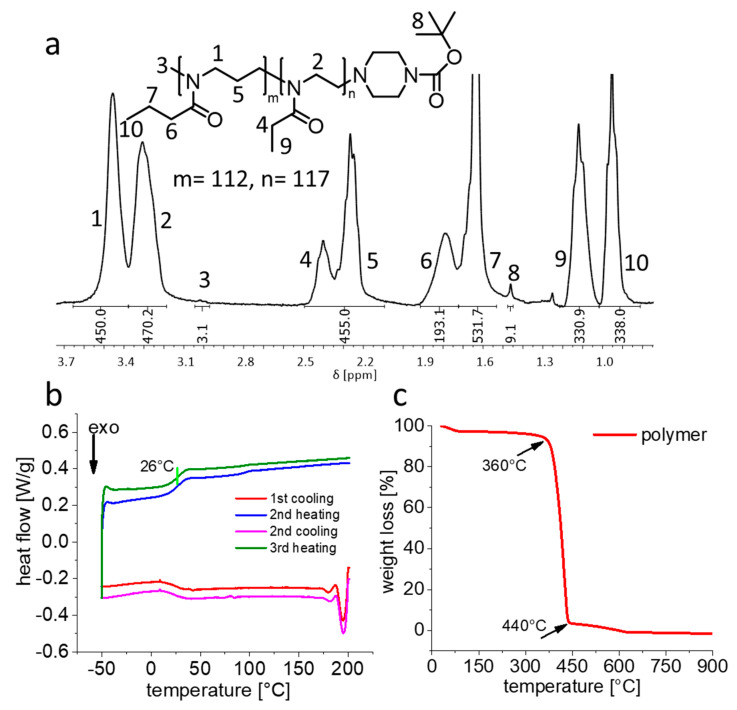
(**a**) Chemical structure and ^1^H-NMR spectra (CDCl_3_, 300 MHz, 298 K) of pPrOzi_100_-*b*-pEtOx_100_ with the signal assignment of all major peaks. (**b**) Heat flow occurring during the several consecutive heating and cooling cycle (10 K/min) of differential scanning calorimetry, green vertical line indicating the glass transition point. (**c**) Weight loss occurring during thermogravimetric analysis of pPrOzi_100_-*b*-pEtOx_100_. Samples were heated from 30 to 900 °C at the heating rate of 10 K/min.

**Figure 3 gels-07-00078-f003:**
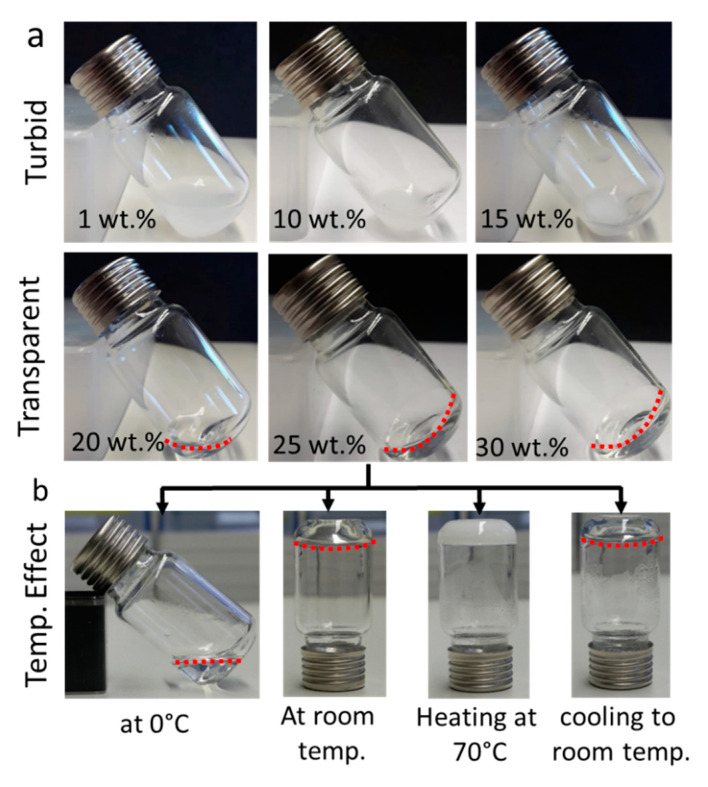
(**a**) Visual appearance of aqueous pPrOzi_100_-*b*-pEtOx_100_ diblock polymer solutions (1 to 30 wt.%) at room temperature. (**b**) Visual appearance of 25 wt.% polymer solution at different temperatures. Dotted red lines are added for better visibility of colourless liquid or hydrogel meniscus.

**Figure 4 gels-07-00078-f004:**
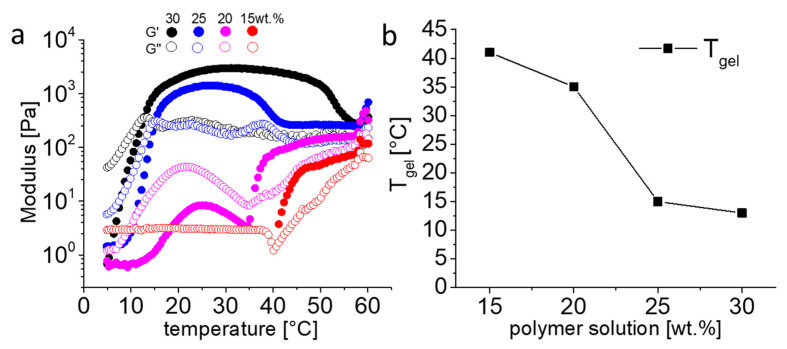
(**a**) Temperature dependent rheology from 5 to 60 °C of the 15, 20, 25 and 30 wt.% diblock copolymer solution with storage modulus (G’) and loss modulus (G”). (**b**) Graphical representation of the reduction in gelation temperature (T_gel_) with the increasing polymer concentration (in wt.%). Lines between data points are guide for the eyes only.

**Figure 5 gels-07-00078-f005:**
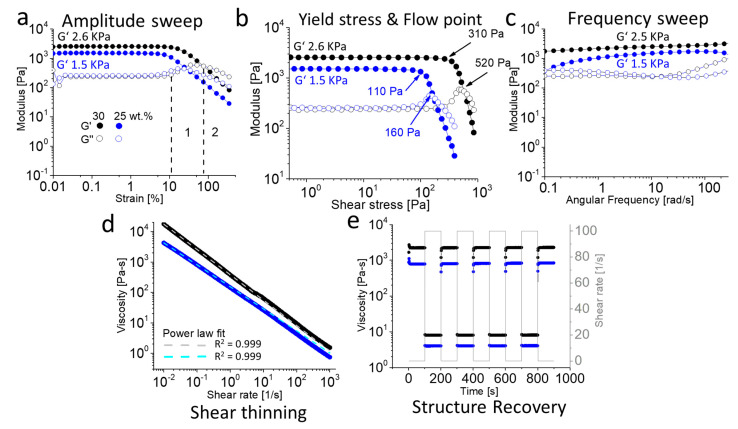
(**a**) Amplitude sweep at an angular frequency of 10 rad/s, zone 1 and 2 (exemplified only for 30 wt.%) indicating the hydrogel’s gradual transition from solid to liquid state under applied strain. (**b**) Yield and flow point values obtained from amplitude sweep (**c**) Frequency sweep at an applied strain of 0.1%. (**d**) Viscosity depending on the applied shear rate at 25 °C representing the shear thinning behaviour including power law fit (dashed lines). (**e**) Structure recovery property under alternative applied shear rate of 0.1 and 100/s. All of the experiments were performed for 25 (blue) and 30 wt.% (black) pPrOzi_100_-*b*-pEtOx_100_ diblock polymer hydrogels.

**Figure 6 gels-07-00078-f006:**
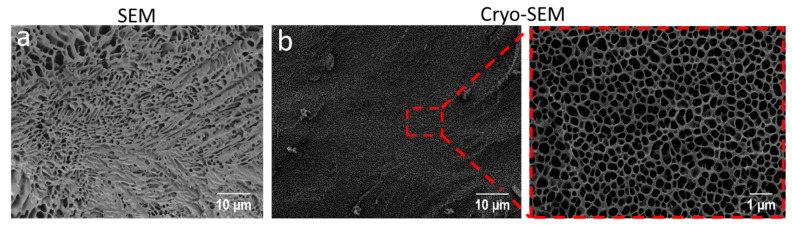
Visualization of the (**a**) morphology of 30 wt.% lyophilized hydrogel by scanning electron microscopy (SEM) and (**b**) freshly prepared hydrogel by cryo-SEM (acceleration voltage ETH: 2 kV).

**Figure 7 gels-07-00078-f007:**
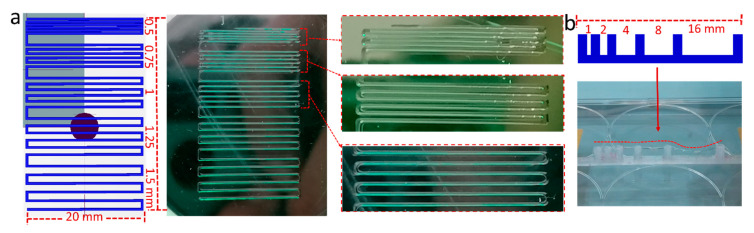
3D-printibility assessment of pPrOzi_100_-*b*-pEtOx_100_ based hydrogel by (**a**) film fusion test (25 wt.%) by printing a stepwise increasing strand to strand distance of 0.5, 0.75, 1.0, 1.25, 1.5 mm (pressure; 80 kPa, speed; 2 mm/s) and (**b**) filament collapse test (30 wt.%) by using a serrated mold of increasing gap (pressure; 120 kPa, speed; 6 mm/s), red dashed line is added for eye guidance. The screen shots of the designed structures were directly taken from Repetier software during G-Code writing.

**Figure 8 gels-07-00078-f008:**
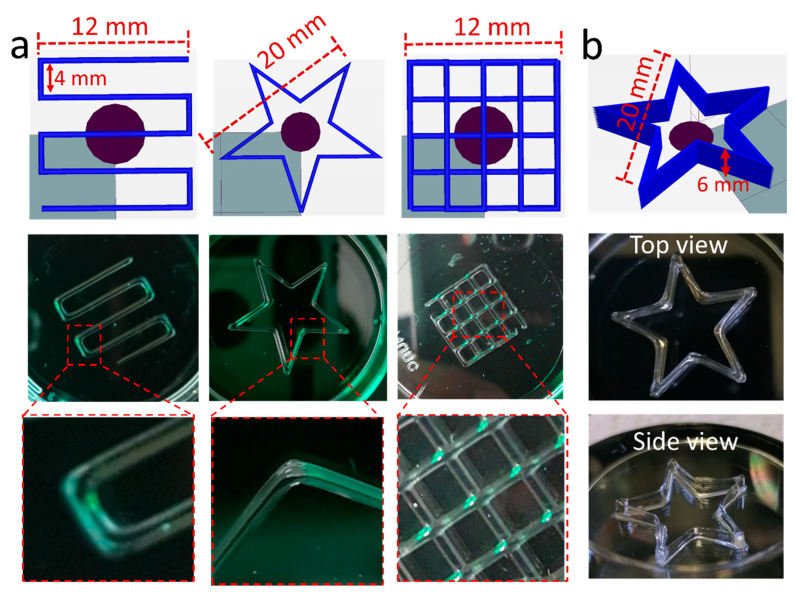
(**a**) 3D printability of 25 wt.% pPrOzi_100_-*b*-pEtOx_100_ based hydrogel with various forms such as serpentine line, a star and a double layer grid (pressure; 90 kPa, speed 2 mm/s) (left to right). (**b**) 3D printability of 25 wt.% pPrOzi_100_-*b*-pEtOx_100_ based hydrogel with 1 wt.% laponite XLG (12 layered star) (pressure; 120 kPa, speed 1 mm/s). The screen shots of the designed structures were directly taken from Repetier software during G-code writing.

**Figure 9 gels-07-00078-f009:**
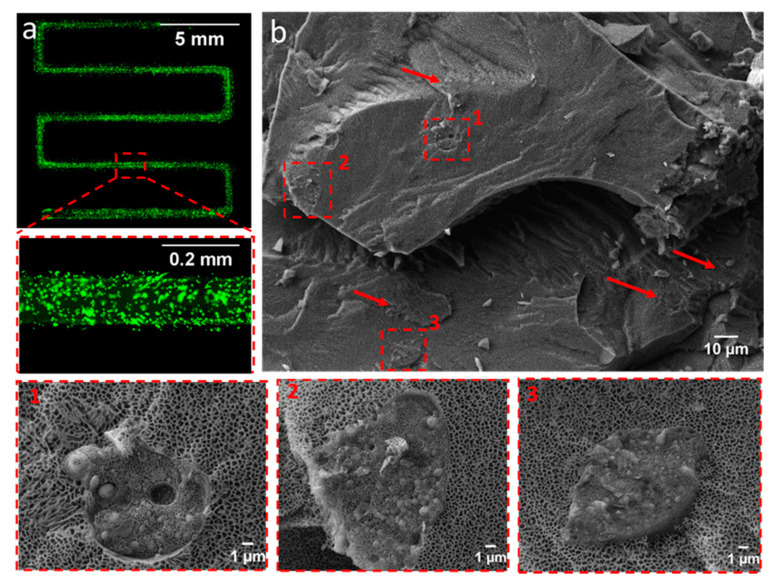
(**a**) Stitched fluorescence microscopic image of the cells distribution in single printed serpentine-like line, hADSCs were labelled with fluorescent dye (**b**) Cryo-SEM image of the cells laden 30 wt.% bio-ink with representative images (1, 2, 3) showing individual cells (arrows are also indicating the presence of cells) (acceleration voltage ETH: 8 kV).

**Figure 10 gels-07-00078-f010:**
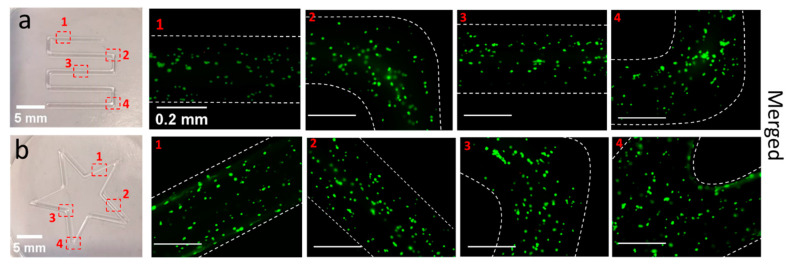
Live (green) and dead (red) viability assay of post-printing cell laden 30 wt.% bio-ink (**a**) serpentine line and (**b**) a star. Number of the live and dead cells were counted from cellular fluorescence images at four different places in individual construct by imageJ software. The images are presented as merged for live and dead cells. Dashed white lines are added for eye guidance (scale bar in fluorescence images is 0.2 mm).

## Data Availability

The data presented in this study are available on request from the corresponding authors.

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
