# Peer review of "Tuning the Thermogelation and Rheology of Poly(2-Oxazoline)/Poly(2-Oxazine)s Based Thermosensitive Hydrogels for 3D Bioprinting"

_gels, 2021, doi:10.3390/gels7030078_

Round 1
Reviewer 1 Report
I have reviewed a manuscript entitled” Tuning the thermogelation and rheology of poly(2-oxazo-line)/poly(2-oxazine)s based thermosensitive hydrogels for 3D 3 bioprinting.” This work developed a synthetic hydrogel with high printability and biocompatibility. I think it is suitable for publication after addressing the following comments:
- Please quantify the rheological behavior of the ink by employing the power-law model. following references might be useful:
https://www.sciencedirect.com/science/article/pii/S0144861721003763
https://pubs.acs.org/doi/10.1021/acsami.7b04216
- Could you please clarify the extent of swelling ratio and degradation kinetics of this hydrogel?
- Is this hydrogel able to print a multi-layered structure? If so, how tall structures can be made by this hydrogel?
- As 1um is small for cell migration and proliferation, what is your suggestion to increase the pore size without compromising the rheological behavior?
Author Response
Response to Reviewer’s comments on Manuscript “gels-1216938” “Tuning the thermogelation and rheology of poly(2-oxazoline)/poly(2-oxazine)s based thermosensitive hydrogels for 3D bioprinting.”
Dear Editor and Reviewers,
Thank you very much for giving us a chance to revise this manuscript. Additionally, we thank the reviewers for critical and professional comments allowing us to improve the quality of the manuscript. We have tried to answer all the comments of both reviewers, please see below.
Reviewer no. 1 Report
I have reviewed a manuscript entitled” Tuning the thermogelation and rheology of poly(2-oxazo-line)/poly(2-oxazine)s based thermosensitive hydrogels for 3D 3 bioprinting.” This work developed a synthetic hydrogel with high printability and biocompatibility. I think it is suitable for publication after addressing the following comments:
Authors Response:
Comment 1: Please quantify the rheological behavior of the ink by employing the power-law model. Following references might be useful: https://www.sciencedirect.com/science/article/pii/S0144861721003763 https://pubs.acs.org/doi/10.1021/acsami.7b04216
Answer: The hydrogel in present contribution has been already characterized by power-law model. The flow indices n and consistency indices K were already given in the main text (please refer to Line 289 to 294) and the power law fit can be seen in Figure 5d. Briefly, the present material exhibited very low flow and rather high consistency indices (n ≈ 0.16 and 0.27, K ≈ 380 and 152, for 30 and 25 wt.% hydrogel, respectively).
No Changes made to the manuscript:
Comment 2: Could you please clarify the extent of swelling ratio and degradation kinetics of this hydrogel?
Answer: We thank the reviewer for this question. Swelling ratio and degradation kinetics were not determined for this hydrogel, as this is physically cross-linked network and do not swell, rather simply dissolve upon dilution.
No Changes made to the manuscript.
Comment 3: Is this hydrogel able to print a multi-layered structure? If so, how tall structures can be made by this hydrogel?
Answer: We thank the reviewer for this question. The stackability of this hydrogels is already presented in Figure 8b and Figure S4 and discussed in the main text (Line 368 to 375). Briefly, with the pristine hydrogel, printing more than three layers leads to slight flattening of the scaffold (Figure S4) while with only 1 wt.% of Laponite XLG (Figure 8b), a 12 layered star was easily printed with good shape fidelity. One would anticipate that stackability can be improved further by increasing the Laponite feed but 12 layers is already a reasonably good performance.
No changes made to the manuscript:
Comment 4: As 1um is small for cell migration and proliferation, what is your suggestion to increase the pore size without compromising the rheological behavior?
Answer: The reviewer correctly noted an important issue of this material. It is clear that pore size is too small to allow cell migration (see Figure 9 and compare size of the embedded cells vs. the gel pore size). In this proof of principle study, we introduce a new gel, which can potentially be used for tissue engineering and biomedical applications. However, as such, it will likely not provide a stand-alone solution. As mentioned in comment 2 response, this is a physically cross-linked hydrogels with good rheology and printability with cells-/survival, at the moment it can not directly be used for longer cell culture, as the scaffold start to dissolve upon dilution. In its current state, it´s most probable user-case would be as sacrificial material, where it can help to improve the printability of poorly printable polymers like alginate or GelMA (see comparable work using a different hydrogel: DOI: 10.26434/chemrxiv.14447676). We anticipate that with certain chemical modifications, the hydrogel could be chemically crosslinked and literature know strategies could be harnessed to control the pore size, may be with the variety of reported simple techniques like solvent casting, freeze drying, gas foaming or with other sophisticated techniques like soft/photo lithography, hydrodynamic focusing or electro-spraying.
No Changes made to the manuscript:
Reviewer 2 Report
Haider and colleagues synthesized thermosensitive pPrOzi100-b-pEtOx100-based diblock copolymers and evaluated their suitability for 3D bioprinting. The work is interesting, but in my opinion some points have to be explained better.
GENERAL COMMENTS
- Why did the Authors perform some investigations with 25% hydrogels and some other investigations with 30% hydrogels? For instance, they reported the shape fidelity for 25% hydrogels, but they embedded and printed the cells in 30% hydrogels. This is confusing. Which is the best choice for 3D bioprinting? If 25% and 30% gels are equally suitable, the Authors have to report the performances of both materials in the text.
- Detail the possible applications of 25% and 30% hydrogels. Claiming that they can be applied in the field of “tissue engineering, regenerative medicine and drug delivery”, as mentioned in the Introduction for smart materials, is very general. The Authors seem to suggest that their final goal is proposing an alternative to PEG. To improve the relevance of the work, this idea has to be better structured and supported with references.
- The Authors did not provide any evidence that the hydrogels are suitable for cell culture. Since they did not mention when they perform live/dead assay, from the text it can only be inferred that 3D printing parameters are suitable for cell survival. Explain better and add references or perform a biocompatibility evaluation (before printing could be sufficient).
- Why did the Authors perform both Scanning Electron Microscopy and Cryo-Scanning Electron Microscopy? Explain why you need to report both techniques (and not only Cryo-Scanning Electron Microscopy) or remove Scanning Electron Microscopy.
OTHER COMMENTS:
- Lines 53-56: Please, rephrase. Several studies have presented PEG as highly cytocompatible, e.g. Li et al., 2012. DOI: 10.1016/j.progpolymsci.2012.02.004. Furthermore, members of the PEG polymer family have been approved by the US Food and Drug Administration as sealants (Burkett et al., 2011; DOI: 10.1016/j.jocn.2011.04.005. Spotnitz and Burks, 2012; DOI: 10.1111/j.1537-2995.2012.03707.x)
- Line 75: “physically more compact”. Rephrase.
- Lines 88-81: “its rheological characteristics, in particular its low yield stress were not conducive for 3D printing”. Clarify. Several techniques exist for 3D printing.
- Line 116: What does “Tcp” mean?
- Lines 125-128: Why did the Authors insert this sentence here? Clarify or remove
- Lines 194-194: These two sentences are unclear. Explain better
- Lines 229-230: The crossover point for G’ and G” was also observed for 20 and 15 wt.% polymer solution at 35 and 41°C, respectively. For 15% hydrogels at temperatures lower than 40°C, do G’ and G” overlap? Data seem to be missing in Figure 4a. Explain
- Lines 259-261: Yield stress… Insert a reference;
- Lines 264-266: In (bio) printing… Insert a reference;
- Line 321:”The filament fusion test was done to assess the shape fidelity” Since the Authors did not use any formulas to evaluate the shape fidelity, I would suggest to define what they mean with “shape fidelity”
- Lines 344-346: The development of multilayered … Insert a reference;
- Lines 427-429: Fibroblasts or adipose-derived stem cells have different features. I do not agree with the idea that adipose-derived stem cells are a routinely used model for fibroblasts. I would suggest to revise the sentences
- “Live (green) and dead (red) viability assay of post-printing cell” What time after printing was it performed? Add in the text
- Lines 484-489: “Additionally..” This is not included in this manuscript. Rephrase the sentence.
- Line 528: What does “HFIP” mean?
- Figures 5A, 5B, 5C: Detail which Modulus. G’? Similarly, in the supplementary figures related to rheology
- Figure 5D: Report the R2 values ofor the Power law fitting
- Figure 7: Add a scale bar. In figures a), the length of the horizontal filaments can not be derived
- Figure 8: Add a scale bar, as in Figures 9 and 10.
- Figure S2a: Check the amplitude sweep for 15% hydrogels. Fir G’ curve, some points are missing
- Figure S4a: This image is unclear. It suggests that the hydrogel is printable, not that a 5-layer structure has been printed. I would suggest showing lateral views, and report a representative image of a 1-layer structure and then an example image of a multilayered structure.
- Figure S7: Add a scale bar and improve the image. Dead cells are not visible
GENERAL COMMENT ABOUT THE REFERENCES:
- The last author is mentioned in about 25% of listed references. It is undeniable that he has published excellent studies in high impact factor journals, but I would suggest reducing the number of self-citations.
Author Response
Reviewer no. 2 Report
Haider and colleagues synthesized thermosensitive pPrOzi100-b-pEtOx100-based diblock copolymers and evaluated their suitability for 3D bioprinting. The work is interesting, but in my opinion some points have to be explained better.
Authors Response:
GENERAL COMMENTS
Comment 1: Why did the Authors perform some investigations with 25% hydrogels and some other investigations with 30% hydrogels? For instance, they reported the shape fidelity for 25% hydrogels, but they embedded and printed the cells in 30% hydrogels. This is confusing. Which is the best choice for 3D bioprinting? If 25% and 30% gels are equally suitable, the Authors have to report the performances of both materials in the text.
Answer: In most cases, in particular the fundamental characterization, both concentrations were investigated. The main considerations for our experimental design was: a) Besides other factors, printability and shape fidelity are directly associated to polymer concentration, and to obtain gel at ambient conditions, 25 wt.% polymer was needed as a minimum. So in our opinion, it was very important to investigate the printability and shape fidelity at 25 wt.% in the first place, and additional experiments at 30 wt.% give insight into the concentration dependency. Interestingly, no significant differences were observed in overall printing performance with the exception of filament collapse test, where 30 wt.% remained continuous while 25 wt.% hydrogel filament collapsed at 16 mm gap distance (Please refer to Line 340 to 353). b) Out of the two concentrations, the 30 wt.% was further processed for cell-laden hydrogel printing, the major reason was to expose the cells to relatively high polymer concentration, high gel stiffness and associated high extrusion pressure. As the cell have withstood all of these stress factors, we can relatively safely assume that cells would tolerate printing at 25 wt.% gel as well, as the conditions should be generally more cell friendly. In contrast, if we would have performed the cell experiments at 25 wt.% gel, extrapolation to 30 wt.% would not be possible. In this way, we have reduce the overall workload and avoided overloading the manuscript with experimental details that would not add much valuable information. There is probably no generally ideal choice for 3D bioprinting, one always has to ask what particular goal one wants to achieve.
Changes made to the manuscript: A statement about hydrogel wt.% used for printing was added to the methods section, Line 668/669.
Comment 2: Detail the possible applications of 25% and 30% hydrogels. Claiming that they can be applied in the field of “tissue engineering, regenerative medicine and drug delivery”, as mentioned in the Introduction for smart materials, is very general. The Authors seem to suggest that their final goal is proposing an alternative to PEG. To improve the relevance of the work, this idea has to be better structured and supported with references.
Answer: We thank the reviewer for pointing out that we could be more clear in this, however, this mentioning of “tissue engineering, regenerative medicine and drug delivery” was written in the first sentence of the introduction, which is quite commonly setting up a more general stage, here in general for “smart” biomaterials, under which framework thermogelling polymers certainly can be counted. Certainly the some of us and others have proposed POx as one potential alternative to PEG, but this study is more on the difference between pMeOx and pEtOx, which both have to suggested as an alternative to PEG. To this end, for example, we wrote in the conclusion: “Although, pMeOx and pEtOx are routinely addressed as alternative and similar material [39, 40, 83] when compared to PEG but it will be interesting to see more of direct comparison of both polymers at molecular level.”. As indicated by title of the manuscript and as discussed in introduction section (line 37 to 42) and results and discussion section, the major aim of this work is to present a platform which can potentially be used for tissue engineering applications. At the same time, study was designed to check the impact of replacing pMeOx with pEtOx, on the rheological and thermogelling properties of resultant diblock copolymer (Please refer to Line 96 to 107 and Line 126 to 132). As mentioned in an answer to reviewer 1 (comment 4), the present material, in its current state, can be used mainly as a sacrificial/support material, where it can help to improve the printability of poorly printable polymers like alginate or GelMA. This possible application has been discussed in main text (please refer to Line 428 to 434). Additionally, we anticipate that with certain chemical modifications, the spectrum of its utility can be increased further (please refer to Line 520 to 525).Additionally, we have summarily discussed the problems with use of PEG as hydrophilic polymer and have emphasized on the need for the development of new alternatives to PEG. Like many other hydrophilic polymers classes, the POx/POzi based hydrophilic polymer also hold the huge potential for further exploration in this regard (please refer to Line 71 to 77).
Changes made to the manuscript: We made some adjustments in the introduction hoping that the manuscript is more clear now.
Comment 3: The Authors did not provide any evidence that the hydrogels are suitable for cell culture. Since they did not mention when they perform live/dead assay, from the text it can only be inferred that 3D printing parameters are suitable for cell survival. Explain better and add references or perform a biocompatibility evaluation (before printing could be sufficient).
Answer: We thank the reviewer for pointing out where we needed to be clearer describing our work. As discussed, the present hydrogel is physically cross-linked network and cannot directly easily be used for long-term cell cultures (exceeding approx. 3 days), as the scaffold would dissolve when fresh medium would be added/replaced (typically necessary after 3 days in static cell culture). Therefore, In the case of post printing live/dead assay, the analysis was done immediately after printing the scaffolds. However, it is worth mentioning here that boink preparation (cells mixing, transfer of bioink to printing cartridge and waiting time to obtain bubble free settled bioink), in general took 45 minutes. Collectively we can say that the whole process took 1 hour from ink preparation to visualization and during this time cell remained viable without any standard culture conditions. At the same time, we could investigate cytocompatibility at concentrations below the critical gel concentration, but, in our opinion, this would not add much scientific value, especially with knowing the fact that POx/POzi have been repetitively established cytocompatible in general and particularly in our lab, e.g. doi.org/10.1021/acs.biomac.7b00481, doi.10.1016/j.jconrel. 2019.04.014. doi.org/10.1039/D0PY01258K. As suggested by the reviewer, we have added additional information hoping to clarify this issue (and related references).
Changes made to the manuscript: The information related to post-printing live and dead assay is updated in results & discussion (Please refer to Line 481 to 488) and in methods section (please refer to Line 688/89).
Comment 4: Why did the Authors perform both Scanning Electron Microscopy (SEM) and Cryo-Scanning Electron Microscopy (cryo-SEM)? Explain why you need to report both techniques (and not only Cryo-Scanning Electron Microscopy) or remove Scanning Electron Microscopy.
Answer: We thank the reviewer for bringing this up. Initially, we had performed only SEM of the lyophilized hydrogels as shown in Figure 6a. It is evident, that the morphology of the hydrogels is affected by the lyophilisation and became a cause of artifacts in SEM analysis. This issue of sample preparation in SEM preparation of hydrogels introducing considerable artifacts has been discussed more and more in recent years. To get more realistic view of the native hydrogels morphology, cryo-SEM was further performed. With cryo-SEM, it was confirmed that lyophilisation indeed strongly affected the hydrogel’s microporous structure. Although this is known and we are certainly not first to notice this, one can still find in many recent papers SEM images of freeze-dried, in which it is claimed that this would show the morphology of the hydrogels. Therefore, in the overall benefit for the scientific community, we would like to keep both analysis techniques so that its underlined once more that sample preparation is critical in this context. The relevant discussion was already present in the main text, section 2.2 SEM and cryo-SEM (Please refer to Line 313 to 325) and a particularly interesting paper by Kaberov et al. was referenced.
No Changes made to the manuscript:
OTHER COMMENTS:
Comment 5: Lines 53-56: Please, rephrase. Several studies have presented PEG as highly cytocompatible, e.g. Li et al., 2012. DOI: 10.1016/j.progpolymsci.2012.02.004. Furthermore, members of the PEG polymer family have been approved by the US Food and Drug Administration as sealants (Burkett et al., 2011; DOI: 10.1016/j.jocn.2011.04.005. Spotnitz and Burks, 2012; DOI: 10.1111/j.1537-2995.2012.03707.x)
Answer: We thank the reviewer for this helpful suggestion. As suggested, the associated references are being added and the associated text is rephrased in introduction section.
Changes made to the manuscript: Please refer to introduction section, Line 55 to 59.
Comment 6: Line 75: “physically more compact”. Rephrase.
Answer: We thank the reviewer for pointing out this.
Changes made to the manuscript: The “physically more compact” is now changed to “Utilizing the light scattering technique, Grube et al. explained the solution properties of POx in comparison to PEG, both pMeOx and pEtOx with same molar mass (as PEG) were less solvated and more compact in shape.” Please refer to introduction section, Line 74 to 77.
Comment 7: Lines 88-81: “its rheological characteristics, in particular its low yield stress were not conducive for 3D printing”. Clarify. Several techniques exist for 3D printing.
Answer: We thank the reviewer for pointing out this. Now the sentence is modified as “its rheological characteristics, in particular its low yield stress were not conducive for extrusion based 3D printing”
Changes made to the manuscript: Please refer to introduction section, Line 93/94.
Comment 8: Line 116: What does “Tcp” mean?
Answer: The “Tcp” is abbreviation for cloud point temperature. The expansion is added to the main text.
Changes made to the manuscript: Please refer to introduction section, Line 119.
Comment 9: Lines 125-128: Why did the Authors insert this sentence here? Clarify or remove
Answer: We thank the reviewer for pointing out this. The reason for adding this (now appearing as Line 126 to 132) is to revise the main aim of the manuscript to the readers, before directly jumping into the synthesis details i.e. although, both pMeOx and pEtOx are hydrophilic polymer, and one can think that these hydrophilic polymer should have the same properties and replacing them with each other would not make a big difference. But actually, this is not the case, previously, we have proved that this small change can significantly impact the drug loading while in present contribution, it has been shown that, the rheological profile and printability is sufficiently affected. For better understanding of the readers, the paragraph is slightly rephrased
Changes made to the manuscript: Please refer to line 126 to 135.
Comment 10: Lines 194-194: These two sentences are unclear. Explain better
Answer: We thank the reviewer for this suggestion. Both of the sentences are rephrased.
Changes made to the manuscript: Please refer to section 2.1, synthesis, characterization and rheology, (now appearing as) Line 199 to 201.
Comment 11: Lines 229-230: The crossover point for G’ and G” was also observed for 20 and 15 wt.% polymer solution at 35 and 41°C, respectively. For 15% hydrogels at temperatures lower than 40°C, do G’ and G” overlap? Data seem to be missing in Figure 4a. Explain
Answer: We thank the reviewer for noticing this. In the temperature sweep, the G’ value for 15 wt.% remained below instrument sensitivity (registers as 0) until 40°C, above this temperature the value start to increase and actually is above G’’. As the log of zero is undefined, no values are plotted in this logarithmic plot. This part was re-written and is hopefully more clear now.
Changes made to the manuscript: Please refer to section 2.1, synthesis, characterization and rheology, Line 238 to 240.
Comment 12: Lines 259-261: Yield stress… Insert a reference;
Answer: We thank the reviewer for this suggestion. The added reference is:
- Balmforth, Neil J., Ian A. Frigaard, and Guillaume Ovarlez. "Yielding to stress: recent developments in viscoplastic fluid mechanics." Annual Review of Fluid Mechanics 46 (2014): 121-146.
Changes made to the manuscript: Please refer to section 2.1, synthesis, characterization and rheology, Line 272.
Comment 13: Lines 264-266: In (bio) printing… Insert a reference;
Answer: We thank the reviewer for this suggestion. The added references are:
- Blaeser, Andreas, et al. "Controlling shear stress in 3D bioprinting is a key factor to balance printing resolution and stem cell integrity." Advanced healthcare materials 5.3 (2016): 326-333.
- Malkoc, Veysi. "Challenges and the future of 3D bioprinting." J Biomed Imaging Bioeng 2.1 (2018): 64-65.
Changes made to the manuscript: Please refer to section 2.1, synthesis, characterization and rheology, Line 277.
Comment 14: Line 321:”The filament fusion test was done to assess the shape fidelity” Since the Authors did not use any formulas to evaluate the shape fidelity, I would suggest to define what they mean with “shape fidelity”
Answer: We thank the reviewer for this valuable suggestion. As suggested the explanation regarding the shape fidelity is included in the main text and associated references is also being cited i.e.
- Schwab, Andrea, et al. "Printability and shape fidelity of bioinks in 3D bioprinting." Chemical Reviews 120.19 (2020): 11028-11055.
Changes made to the manuscript: Please refer to section 2.3, 3D-printing of hydrogels, Line 338 to 343
Comment 15: Lines 344-346: The development of multilayered … Insert a reference;
Answer: We thank the reviewer for this suggestion. The associated reference is added i.e.
- Zhuang, Pei, et al. "Layer-by-layer ultraviolet assisted extrusion-based (UAE) bioprinting of hydrogel constructs with high aspect ratio for soft tissue engineering applications." PloS one 14.6 (2019): e0216776.
Changes made to the manuscript: Please refer to section 2.4, Printability and Rheology, Line 369.
Comment 16: Lines 427-429: Fibroblasts or adipose-derived stem cells have different features. I do not agree with the idea that adipose-derived stem cells are a routinely used model for fibroblasts. I would suggest to revise the sentences
Answer: It appears our formulation led to a misunderstanding. We did not intend to say that that adipose-derived stem cells are a routinely used model for fibroblasts, which they are certainly not as the reviewer correctly mentions. Rather, we intended to write that instead of fibroblast, which are relatively robust are commonly used for preliminary testing, while for applications other cells type, including stem cells are needed. For clarification, the whole paragraph is rephrased in the main text.
Changes made to the manuscript: Please refer to section 2.5, Biological studies, Line 456 to 459.
Comment 17: “Live (green) and dead (red) viability assay of post-printing cell” What time after printing was it performed? Add in the text
Answer: We thank the reviewer for this suggestion. Here our answer is similar to the answer given for comment no. 3. Summarily, the live and dead assay was performed immediately after printing the bioink. However, it is worth mentioning here that boink preparation (cells mixing, transfer of bioink to printing cartridge and waiting time to obtain bubble free settled bioink), in general took 45 minutes. Collectively we can say that the whole process took 1 hour from ink preparation to visualization and during this time cell remained viable without any standard culture conditions.
Changes made to the manuscript: The information related to post-printing live and dead assay is updated in results & discussion (Please refer to Line 481 to 488) and in methods section (please refer to Line 688/89).
Comment 18: Lines 484-489: “Additionally..” This is not included in this manuscript. Rephrase the sentence.
Answer: We thank the reviewer for this suggestion. The “additionally” is replaced with “we anticipate”, and the whole sentence is further rephrased.
Changes made to the manuscript: Please refer to section 3, conclusion, Line 520 to 523.
Comment 19: Line 528: What does “HFIP” mean?
Answer: The HFIP stands for Hexafluoro isopropanol. The full name is added in the Methods section.
Changes made to the manuscript: Please refer to section 5.3, Gel permeation chromatography, Line 570.
Comment 20: Figures 5A, 5B, 5C: Detail which Modulus. G’? Similarly, in the supplementary figures related to rheology
Answer: We thank the reviewer for noticing this detail. The Modulus. G’ values are being added and figures are updated accordingly. The relevant discussion is also added to the main text.
Changes made to the manuscript: Please refer to Figure 5 a, b, c in main text and Figure S5 and Figure S6 in the supplementary information. The relevant discussion for Figure S6 is also added to the main text (Please refer to section 2.5 biological studies, Line 449 to 452).
Comment 21: Figure 5D: Report the R2 values for the Power law fitting
Answer: We thank the reviewer for this suggestion. The R2 values are being added and Figure 5d is updated accordingly.
Changes made to the manuscript: Please see figure 5d in main text.
Comment 22: Figure 7: Add a scale bar. In figures a), the length of the horizontal filaments can not be derived
Answer: We thank the reviewer for this suggestion. The Figure 7a (G-code screen shot) is updated with horizontal filament dimension (i.e. 20 mm) of construct, now the length of horizontal filament is clear and easily derived.
Changes made to the manuscript: Please see Figure 7a (G-code screen shot) in main text.
Comment 23: Figure 8: Add a scale bar, as in Figures 9 and 10.
Answer: We thank the reviewer for this suggestion. The dimensions of the constructs are already given in the Repetier G-Code screen shots (Figure 8, top panel). However, further details are added now i.e. Figure 8a, 4mm (width) and Figure 8b, 6mm (Z-axis height). We hope that it is now much clearer to derive the dimensions of original printed constructs.
Changes made to the manuscript: Please see Figure 8, top panel.
Comment 24: Figure S2a: Check the amplitude sweep for 15% hydrogels. For G’ curve, some points are missing.
Answer: We thank the reviewer for this observation and suggestion. Here, the explanation is similar to comment No. 11. In the amplitude sweep, the G’ value for 15 wt.% remained zero at certain strain levels, therefore because of logarithmic plot, it is not possible to see the values with zero registration.
No Changes made to the manuscript:
Comment 25: Figure S4a: This image is unclear. It suggests that the hydrogel is printable, not that a 5-layer structure has been printed. I would suggest showing lateral views, and report a representative image of a 1-layer structure and then an example image of a multilayered structure.
Answer: We agree with the reviewer in his point. The lateral views of both single and five-layered star has been added. The Figure S4 and its caption is updated accordingly.
Changes made to the manuscript: Please see figure S4 in supporting information.
Comment 26: Figure S7: Add a scale bar and improve the image. Dead cells are not visible
Answer: We thank the reviewer for this suggestion. The scale bar is added and the quality of image is improved and the dead (red) cells are relatively better visible now.
Changes made to the manuscript: Please see Figure S7 in supporting information.
GENERAL COMMENT ABOUT THE REFERENCES:
Comment 27: The last author is mentioned in about 25% of listed references. It is undeniable that he has published excellent studies in high impact factor journals, but I would suggest reducing the number of self-citations.
Answer: We thank the reviewer for this compliment and critique. We do try not to have excessive self-citations. A number of the self citations were removed and references by others added. The manuscript now stands at 19%, which we believe is agreeable and we hope the reviewer agrees.
Changes made to the manuscript: Self citations are reduced from 25 to 19%.
Round 2
Reviewer 1 Report
I think it can be considered for publication.
Reviewer 2 Report
I would like to thank the Authors for providing a new, clearer, version of their manuscript.